# *Lactobacillus delbrueckii* Ameliorated Blood Lipids via Intestinal Microbiota Modulation and Fecal Bile Acid Excretion in a Ningxiang Pig Model

**DOI:** 10.3390/ani14121801

**Published:** 2024-06-17

**Authors:** Gaifeng Hou, Liangkai Wei, Rui Li, Fengming Chen, Jie Yin, Xingguo Huang, Yulong Yin

**Affiliations:** 1Key Laboratory of Agro-Ecological Processes in Subtropical Region, Institute of Subtropical Agriculture, Chinese Academy of Sciences, Changsha 410125, China; hougf521@163.com (G.H.); yinyulong@isa.ac.cn (Y.Y.); 2Hunan Co-Innovation Center of Safety Animal Production, College of Animal Science and Technology, Hunan Agricultural University, Changsha 410128, China; wlk20210730@163.com (L.W.); yinjie2014@126.com (J.Y.); hxg68989@hunau.edu.cn (X.H.); 3Academician Workstation, Changsha Medical University, Changsha 410219, China; cfming@csmu.edu.cn

**Keywords:** *Lactobacillus delbrueckii*, fat metabolism, intestinal microbiota, Ningxiang pigs

## Abstract

**Simple Summary:**

Ningxiang pigs are popular for their high meat quality and unique flavor, but they contain a low lean percentage. Long-term excessive intake of pork with high fat content and its products might be associated with dyslipidemia. *Lactobacillus delbrueckii* has been proven to regulate lipid metabolism and improve blood lipids in our previous studies; however, the underlying mechanism remains unclear. In the present study, we investigated the effects of *L. delbrueckii* on gut microbiota and body lipid metabolism and found that *L. delbrueckii* regulated hepatic and tissular fat metabolism via gut microbiota modulation and fecal TBA excretion to reduce blood lipid levels in Ningxiang pigs.

**Abstract:**

*Lactobacillus delbrueckii* intervention can regulate body lipid metabolism, but the underlying mechanism remains unclear. Our study investigated the effects of *L. delbrueckii* on serum lipid levels, tissular fat metabolism and deposition, bile acid metabolism, and gut microbiota in Ningxiang pigs. Ninety-six pigs were divided into two groups and fed basal diets containing either 0 (CON) or 0.1% *L. delbrueckii* (LD) for 60 days. Dietary *L. delbrueckii* promoted fecal total bile acid (TBA) excretion and increased hepatic enzyme activities related to cholesterol and bile synthesis but decreased hepatic and serum lipid concentrations. *L. delbrueckii* downregulated gene expression associated with fatty acid synthesis but upregulated gene expression related to lipolysis and β-fatty acid oxidation in liver and subcutaneous fat. *L. delbrueckii* elevated gut *Lactobacillus* abundance and colonic short-chain fatty acid (SCFA)-producing bacteria but declined the abundance of some pathogenic bacteria. These findings demonstrated that *L. delbrueckii* modulated intestinal microbiota composition and facilitated fecal TBA excretion to regulate hepatic fat metabolism, which resulted in less lipid deposition in the liver and reduced levels of serum lipids.

## 1. Introduction

The blood lipid levels in the Chinese population have gradually increased over the past three decades [1]. Hyperlipidemia is characterized by high levels of TC, LDL-C, and TG but low levels of HDL-C in the blood [2,3]. A report from the 2016 Chinese guidelines for the management of dyslipidemia in adults showed that the average contents of serum TG and TC were 1.38 mmol/L and 4.5 mmol/L, and the prevalence of hypertriglyceridemia and hypercholesterolemia were 13.1% and 4.9% in adults, respectively [4]. Dyslipidemia is a major causative factor for atherosclerotic cardiovascular disease (ASCVD) [5], which has become the leading cause of death worldwide [6]. An epidemiologic survey revealed that the prevalence of dyslipidemia reached up to 34.0% in Chinese individuals [7]. Eating habits and lifestyle are closely associated with human health and disease. Chronic excessive intake of energy, fat, and cholesterol and a lack of physical exercise easily result in excess weight, obesity, diabetes, and ASCVDs, usually accompanied by dyslipidemia [8,9].

Pork is one of the most important animal foods for Chinese people, accounting for over 60% of national meat consumption and contributing to approximately 50% of global pork output [9]. However, long-term excessive intake of pork with high fat content and its products might be associated with dyslipidemia due to overconsumption of energy and fat, resulting in hyperlipidemia, cardiovascular disease, and type 2 diabetes, usually characterized by high levels of blood total cholesterol and triglyceride [9,10]. Therefore, clarifying the mechanisms of fat metabolism in pigs and developing targeted regulatory strategies for lipid metabolism have promising implications for pork production, which might help control dyslipidemia via dietary interventions. Additionally, given the high similarity in anatomy, physiology, polyphagy, habits, and metabolism, swine may be a good model for studying human nutrient metabolism and health [11].

Ningxiang pigs, a local fat-type breed with less than 35% lean percent, are popular for their tender and succulent meat, unique flavor, higher intramuscular fat (IMF) content, and high-quality unsaturated fatty acid content compared to imported commercial pigs (lean type) [12,13]. Lean and fat pigs with distinct genetic characteristics, as indicated by great differences in fat deposition, are valuable models for exploring the mechanisms of fat metabolism [14]. The lipid-lowering effects of *Lactobacillus delbrueckii* in lean pigs have been confirmed in our previous studies [3,9,15]; however, the role of the strain in lard-type pigs is unclear. In this study, we investigated the effects of *L. delbrueckii* on intestinal microbiota composition, hepatic fat metabolism, tissue fat deposition, and blood lipid levels in Ningxiang pigs.

## 2. Materials and Methods

### 2.1. Bacterial Strain

*Lactobacillus delbrueckii* (Storage No.: M207096) was acquired from the microbiology functional laboratory at the College of Animal Science and Technology at the Hunan Agricultural University (Changsha, China). The strain was activated in MRS (de Man, Rogosa, and Sharpe) medium and sent to the PERFLY-BIO (Changsha, China) for scale-up production, and the detected viable counts of final products reached 5 × 10^11^ CFU/g.

### 2.2. Animals, Diets and Experimental Design

Ninety-six healthy Ningxiang pigs with similar age or initial body weight (110 ± 2 d, 33.28 ± 1.15 kg) were randomly assigned to 2 groups with 6 pens, and each pen had 8 pigs (the same ratio of female to male). Pigs were provided basal diets supplemented with either 0 (CON) or 0.1% *L. delbrueckii* (5 × 10^10^ CFU/g, LD) for 60 days. The basal diets (Table 1) were formulated according to the standard of NY/T 65-2004 to meet or exceed the requirement for 30–50 kg and 50–80 kg pigs, respectively [16]. All pigs were fed their respective diets twice daily (8:00 a.m. and 15:00 p.m.) ad libitum, and water was available all the time. Pigs in each pen were weighed together at the start and end of each period, and the daily feed consumption per pen was recorded. Fecal samples from each pen were collected for 3 consecutive days before the conclusion of the feeding experiment. On day 61, one pig per pen (3 male and 3 female pigs per group) with similar body weight was selected to collect anterior venous blood and then slaughtered by electric shock to isolate the gut, liver, subcutaneous fat, and *Longissimus dorsi*.

### 2.3. Sample Collection and Preparation

Serum was obtained, aliquoted, and stored at −20 °C for lipid analysis. Digesta (in the ileum and colon) and tissues (in the liver, *Longissimus dorsi*, and subcutaneous fat) were quickly removed, snap-frozen in liquid nitrogen, and stored at −80 °C for microbiota composition, lipid, enzyme activity, and gene mRNA expression measurements.

### 2.4. Determination of Lipid Contents in Serum and Tissue

The concentrations of serum triglyceride (TG), total cholesterol (TC), low-density lipoprotein cholesterol (LDL-C), and high-density lipoprotein cholesterol (HDL-C) were determined by the BS 200 automatic blood biochemical analyzer (Mindray, Shenzhen, China) with corresponding kits.

The total protein contents (g protein/L) in the liver, *Longissimus dorsi*, and subcutaneous fat were quantified using BCA protein assay reagent kits (Nanjing Jiancheng Bioengineering Institute, Nanjing, China). About 100 mg of selected tissues were homogenized with 1 mL of a chloroform/methanol solution (2:1, *v*/*v*). The homogenate was centrifuged at 3000 r/min for 10 min at 4 °C to extract tissue lipids. The contents of TC (mmol/g.prot), TG (mmol/g.prot), and TBA (μmol/g·protein) in the selected tissue were measured by corresponding commercial kits (Nanjing Jianchen Bioengineering Institute, Nanjing, Jiangsu, China).

### 2.5. Determination of Enzyme Activity Related to Fat Metabolism

Hepatic total protein contents (g protein/L) were quantified using a BCA protein assay reagent kit (Nanjing Jiancheng Bioengineering Institute, Nanjing, Jiangsu, China). The contents of hepatic fatty acid synthetase (FAS, U/g.prot), adipose triglyceride lipase (ATGL, U/g.prot), β-hydroxy-β-methylglutarate monoyl coenzyme A reductase (HMGR, U/g.prot), and cholesterol-7α hydroxylase (CYP7A1, U/g.prot) were detected using corresponding commercial ELISA kits (Jiangsu Meimian Industrial Co., Ltd., Yancheng, Jiangsu, China).

### 2.6. RT-PCR to Determine the mRNA Expression of Genes Related to Fat Metabolism

The total RNA of tissues in the liver and subcutaneous fat was isolated and reverse-transcribed to cDNA as previously described [9]. The two-step qRT-PCR reactions were performed in triplicate on 96-well plates using a 7500 real-time PCR system (Applied Biosystems, Foster, CA, USA) with the SYBR^®^ Premix Ex Taq^TM^ (TaKaRa Biotechnology (Dalian), China). The primer sequences (Table 2) for FAS, lipoprotein lipase (*LPL*), carnitine palmitoyltransferase (*CPT*1), peroxidase proliferative receptor *γ* (*PPARγ*), and glyceraldehyde-3-phosphate dehydrogenase (*GAPDH*) were synthesized by Sangon Biotech (Shanghai, China). Target gene expression was calculated using the 2^−ΔΔt^ method relative to *GAPDH* gene amplification.

### 2.7. Fecal TC and TBA Content Determination

Fecal lipids were extracted as described above for TC analysis. Fecal TBA was extracted according to our previous description [3]. Briefly, 1 g of frozen fecal sample was dissolved in 40 mL of methanol. After 4 min of sonication and 1 h of shock, the mixture was centrifuged at 10,000× *g* for 10 min to collect the supernatants. Total TC (mmol/L) and TBA (μmol/L) concentrations in the supernatants were determined using a commercial kit purchased from the Nanjing Jianchen Bioengineering Institute. Finally, fecal TC and TBA contents (mg/g) were obtained by formula conversion.

### 2.8. Intestinal Bacterial Composition and Structure

Microbiota composition and structure in ileal and colonic digesta were analyzed via 16S rDNA sequencing, as described in our previous study [3]. Briefly, total DNA was extracted and purified from ileal and colonic digesta samples (*n* = 4 pigs/group) using a TIANamp Stool DNA kit (Tiangen Biotech (Beijing) Co., Ltd., China). DNA quality and quantity were evaluated by gel electrophoresis and a NanoDrop ND-1000 spectrophotometer (Thermo Fisher Scientific, Waltham, MA, USA), respectively. Finally, eight acceptable DNA samples were delivered to LC-Bio (Hangzhou, China) for 16S rDNA sequencing.

The V3-V4 hypervariable region of the bacterial 16S rDNA gene was amplified with barcoded universal primers (341F-806R). Purified amplicons were sequenced on the Illumina HiSeq platform (Illumina, San Diego, CA, USA) according to the standard procedures in LC-Bio (Hangzhou, China). Sequences with 97% similarity were assigned to the same operational taxonomic units (OTUs). An OTU table was further generated to record the abundance of each OTU in each sample, and a profiling histogram was made using R software (v3.1.1) to represent the relative abundance of taxonomic groups from phylum to species. A Venn diagram was generated to visualize the occurrence of shared and unique OTUs among groups.

### 2.9. Analysis of SCFA Composition in Gut Contents

The concentrations of acetic acid, propionic acid, and butyric acid in ileal and colonic digesta were determined using an Agilent 7890A gas chromatographer (Agilent Technologies Inc., Palo Alto, CA, USA) according to our previous method [9,17].

### 2.10. Statistics Analysis

All results were expressed as mean ± SD. Statistical analysis was conducted by the two-tailed unpaired Student’s t-test of SPSS 17.0 (SPSS Inc., Chicago, IL, USA), with an individual pig as an experimental unit. The Ribosomal Database Project (RDP, V.11.3), Greengenes (V.13_8), and NCBI 16SMicrobial were used for data analysis, comparison, and annotation of 16S rDNA sequences. Shapiro–Wilk’s test was used to check the normal distribution of the data. The Kruskal test was used for post hoc comparisons of taxonomies. Pearson correlation analysis between the differential microbiota and measured SCFAs was conducted by GraphPad Prism 7.0 (GraphPad Software Inc., San Diego, CA, USA). For all tests, *p* < 0.05 was considered a significant difference, while 0.05 < *p* < 0.10 was considered a tendency.

## 3. Results

### 3.1. L. delbrueckii did Not Affect the Growth Performance

Compared with the CON group, Dietary *L. delbrueckii* did not affect the average daily weight (ADG, 643 g vs. 658 g) or average daily feed intake (ADFI, 2382 g vs. 2389 g) of Ningxaing pigs.

### 3.2. L. delbrueckii Reduced Serum Lipid Levels

Serum TG, TC, and LDL-C contents in LD were significantly decreased (Figure 1A) relative to those in CON. However, the serum HDL-C/LDL-C in LD was notably increased (Figure 1B) compared with CON.

### 3.3. L. delbrueckii Altered Lipid Contents in Liver

Dietary *L. delbrueckii* remarkably reduced the hepatic concentrations of TG and TC (Figure 2A) but did not change the TG and TC contents in *Longissimus dorsi* and subcutaneous fat (Figure 2A) or hepatic TBA content (Figure 2B).

### 3.4. L. delbrueckii Affected Hepatic Cholesterol and Bile Acid Metabolism

Compared with CON, hepatic HMGR activity tended to elevate in LD (Figure 3A, *p* = 0.092). The increased activity of hepatic CYP7A1 appeared in LD (Figure 3A), but no differences in the hepatic activities of FAS and ATGL were observed between the two groups (Figure 3B,C).

### 3.5. L. delbrueckii Regulated Gene Expression Related to Fatty Acid Synthesis, Lipolysis and β-oxidation in Liver and Subcutaneous Fat

Hepatic mRNA expression of *FAS* tended to downregulate (*p* = 0.076), but *LPL* and CPT-1 were significantly upregulated by LD (Figure 4A). The upward trend of CPT-1 (*p* = 0.072) and increased expression of LPL in subcutaneous fat were found in LD (Figure 4B).

### 3.6. L. delbrueckii Facilitated Fecal TBA Excretion

Dietary *L. delbrueckii* notably increased fecal TBA concentrations but did not affect fecal TC contents (Figure 5).

### 3.7. L. delbrueckii Altered Intestinal Microbiota Structure

The microbiota structure in the ileal digesta was analyzed by 16S rDNA sequencing (Figure 6). The Venn diagram identified 189 shared OTUs between the two groups, and 120 and 81 unique OTUs were observed in the CON and LD groups, respectively (Figure 6A). The α-diversity analysis showed that the observed species (Figure 6B) and Chao1 (Figure 6E) indices were markedly reduced in LD in contrast to CON. The dominant phyla present in ileal digesta were Firmicutes and Proteobacteria, accounting for more than 99% of the total OTUs (Figure 6F and Appendix A), and dietary treatment did not affect ileal bacterial taxa at the phylum level. At the genus level, the relative abundances of *Pasteurella*, *Actinobacillus*, *Haemophilus*, *Streptococcus*, *Veillonella*, and *Cellulosilyticum* were significantly decreased (Figure 6G and Appendix A), whereas the relative abundance of *Lactobacillus* was markedly increased.

The microbiota analysis of colonic digesta is presented in Figure 7. The Venn diagram displayed 182 unique OTUs in CON, 219 unique OTUs in LD, and 1376 shared OTUs in the two groups (Figure 7A). The α-diversity indices of colonic bacteria were not different between the two groups (Figure 7B–E). At the phylum level, Firmicutes, Bacteroidetes, Spirochaetes, and Actinobacteria were the four dominant bacterial populations, accounting for more than 98% of total OTUs (Figure 7F and Appendix A). The Tenericutes and WPS-2 abundances in LD were significantly reduced relative to those in CON. At the genus level, compared with CON, LD notably increased *Butyricicoccus* abundance but decreased *Anaerovibrio* abundance (Figure 7G and Appendix A). The abundances of *Lactobacillus* (*p* = 0.065), *Streptococcus* (*p* = 0.073), and *Doreaabundance* (*p* = 0.087) tended to increase, while the abundances of *Eubacterium* (*p* = 0.096), *Clostridium* IV (*p* = 0.056), and *Ruminococcus*2 (*p* = 0.065) had a downtrend in LD.

### 3.8. L. delbrueckii Changes Intestinal SCFA Profiles

Compared with the CON group, the ileal contents of acetic acid decreased, and the total SCFAs tended to decline in the LD group (Table 3). Higher butyric acid contents were observed in the LD group than in the CON group, and the colonic concentrations of propionic acid and total SCFAs tended to increase in the LD group relative to the CON group.

### 3.9. Correlation between Intestinal Microbiota and SCFA Contents

The *Pasteurella* abundance had a negative correlation with acetic acid content in the ileal digesta (Figure 8A), whereas the *Cellulosilyticum* abundance was positively associated with the concentrations of acetic acid, propionic acid, and total SCFAs. There was a positive relationship between colonic *Anaerovibrio* and propionic acid (Figure 8B), while a negative correlation was observed between Eubacterium and propionic acid or total SCFAs. Meanwhile, *Butyricicoccus* had a positive association with acetic or butyric acid.

## 4. Discussion

Blood lipids are closely associated with human health. Long-term high TG, TC, and LDL-C levels, along with low HDL-C levels, characterize hyperlipidemia, inducing cardiovascular disease (CVD) [3,18]. Probiotic intervention is a potential strategy for the prevention and treatment of hyperlipidemia. *Lactobacillus*, a common probiotic, has been extensively studied and used in humans and animals. Some reviews on the lipid control of *Lactobacillus* demonstrated that the consumption of *Lactobacillus* exerted beneficial effects on the improvement of serum lipids [19,20]. In this study, dietary *L. delbrueckii* lowered the serum TG, TC, and LDL-C concentrations in Ningxiang pigs, coinciding with our previous results [3,9,15]. Our recent findings indicated that the serum lipid-lowering effects of *L. delbrueckii* in pigs were primarily mediated by its bile salt hydrolase (BSH) activity, intestinal microbiota modulation, and SCFA-mediated fat metabolism [3,9].

The swine intestine hosts a vast, complex, and dynamic microbial community, exhibiting a vital role in host metabolism and health [21,22]. The marked shifts in the relative abundance of gut microbiota from the phylum to the genus level are closely bound up with various host physiological statuses [21]. *Pasteurella*, *Actinobacillus*, *Haemophilus*, and *Streptococcus* are prevalent bacterial pathogens associated with respiratory tract infections in the swine industry [23,24,25], resulting in inflammation and impaired growth performance [26]. *Veillonella* is tightly linked to intestinal dysbiosis and disease status, leading to inflammation [27]. Interestingly, we found that the ileal abundances of *Pasteurella*, *Actinobacillus*, *Haemophilus*, *Streptococcus*, and *Veillonella* were lowered by *L. delbrueckii* addition. Additionally, consistent with our previous studies [3,9], greater ileal *Lactobacillus* abundance appeared in the LD group, possibly a cause of the reduction in pathogenic bacteria due to competitive inhibition. These findings suggested that dietary *L. delbrueckii* maintained and improved foregut health by lowering the abundance of some pathogenic bacteria, which provided an explanation for the growth promotion of Ningxiang pigs in previous experiments [28]. The abundance of *Cellulosilyticum* was positively related to pig growth due to the characteristics of cellulose and starch decomposition [29,30]. Surprisingly, the ileal abundance of *Cellulosilyticum* was decreased by *L. delbrueckii*, likely due to fewer substrates in the ileum for the strain to use because of the increased foregut digestibility. Additionally, the reduced concentrations of ileal acetic acid and total SCFAs in the study also indicated that ileal fermentation was weakened due to a lack of substrates.

The hindgut is the main site for fermentation, and undigested nutrients in the distal ileum will flow into the large bowel for microbial fermentation [17,31]. Most SCFAs are generated in colon fermentation, and more than 95% of them are rapidly absorbed and transported into the blood and subsequently taken up and metabolized by body organs as substrates or signal molecules, participating in energy and lipid metabolism [9]. *Anaerovibrio*, *Butyricicoccus*, *Eubacterium*, *Clostridium* IV, and *Ruminococcus*2 are SCFA-producing bacteria. We observed that dietary *L. delbrueckii* increased colonic butyric acid and SCFA contents but reduced colonic propionic acid. Butyric acid is a preferred energy substrate for colonocytes and is almost fully used onsite, whereas acetate and propionate move to the liver via the portal vein for cholesterol and fatty acid synthesis and gluconeogenesis. We also found that dietary *L. delbrueckii* elevated *Butyricicoccus*, *Lactobacillus*, *Streptococcus*, and *Dorea* abundance but declined *Anaerovibrio*, *Eubacterium*, *Clostridium* IV, and *Ruminococcus*2 abundance. *Butyricicoccus* is a common butyrate-producing bacteria [32], and *Anaerovibrio* is known to produce propionate as a major fermentation product [33]. *Eubacterium* was positively linked with visceral fat mass in people [34]. *Clostridium* IV and *Ruminococcus*2 are closely related to obesity caused by excessive calorie intake and blood glucose [35,36]. Additionally, we found that the Firmicutes abundance and F/B ratio were numerically increased, while the abundance of Bacteroidetes was numerically reduced by *L. delbrueckii* in the colon, which was negatively associated with obesity [9]. The above results indicated that *L. delbrueckii* affects host fat and energy metabolism by altering gut microbiota composition and metabolites.

The liver is the center of lipid synthesis and metabolism in the body, playing a key role in maintaining the dynamic balance of blood lipids. HMGR and CYP7A1 are key enzymes for cholesterol and bile acid synthesis in the liver, respectively [37]. Our results showed that *L. delbrueckii* increased hepatic HMGR and CYP7A1 activities, suggesting that the synthesis of cholesterol and bile acid might be enhanced. However, the hepatic TC content was reduced in the LD group, possibly due to cholesterol conversion to bile acids. Cholesterol is a precursor to bile acid synthesis, and increased bile acid synthesis requires more cholesterol, which in turn enhances cholesterol synthesis [3,38]. Meanwhile, dietary *L. delbrueckii* accelerated fecal bile acid excretion, suggesting that fewer bile acids returned to the liver via enterohepatic circulation. As a result, hepatic bile acid synthesis was boosted to compensate for the loss of enterohepatic circulation. This result also provides an explanation for the enhanced hepatic HMGR and CYP7A1 activities in Ningxiang pigs. FAS and ATGL are key enzymes in fatty acid synthesis and decomposition, respectively [39]. Dietary *L. delbrueckii* did not affect the activities of FAS and ATGL in the liver but downregulated hepatic *FAS* expression and upregulated hepatic *CPT*-1 and *LPL* expression. FAS is responsible for the de novo synthesis of fatty acids. Although the mRNA expression of *FAS* was downregulated, its enzyme activity did not change, likely not affecting fatty acid synthesis. Curiously, hepatic TC concentrations were reduced by *L. delbrueckii*, which might be due to enhanced lipolysis. LPL is in charge of lipolysis, and CPT-1 is a key enzyme in fatty acid *β*-oxidation [40]. We observed that *CPT*-1 and *LPL* expression were increased in the liver and subcutaneous fat, suggesting that *L. delbrueckii* supplementation might promote lipolysis, which adds new evidence for the lipid control of *Lactobacillus*.

## 5. Conclusions

*Lactobacillus delbrueckii* regulates hepatic and tissular fat metabolism via intestinal microbiota modulation and fecal bile acid excretion to reduce serum TC and TG levels in Ningxiang pigs (Appendix A).

## Figures and Tables

**Figure 1 animals-14-01801-f001:**
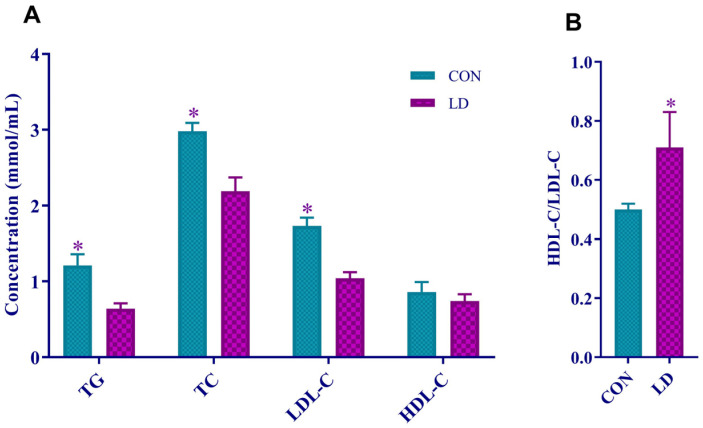
*L. delbrueckii* reduced serum lipid levels. Serum levels of TG, TC, LDL-C, and HDL-C (**A**) and the ratio of HDL-C to LDL-C (**B**) in Ningxaing pigs. Values in columns are shown as the mean ± SD, * *p* < 0.05. TG = triglyceride; TC = total cholesterol; LDL-C = low-density lipoprotein cholesterol; HDL-C = high-density lipoprotein cholesterol; CON = basal diets; LD = basal diets + 0.1% *L. delbrueckii*.

**Figure 2 animals-14-01801-f002:**
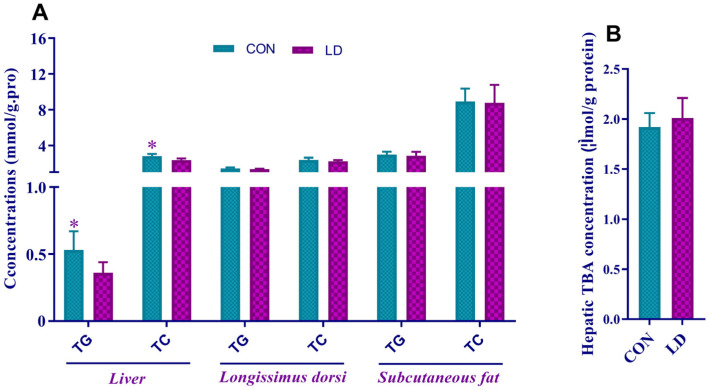
*L. delbrueckii* altered lipid contents in the liver but not in other selected tissues. Concentrations of TG and TC (**A**) and TBA (**B**) in the selected tissues of Ningxiang pigs. Values in columns are shown as the mean ± SD, * *p* < 0.05. TG = triglyceride; TC = total cholesterol; TBA = total bile acid; CON = basal diets; LD = basal diets + 0.1% *L. delbrueckii*.

**Figure 3 animals-14-01801-f003:**
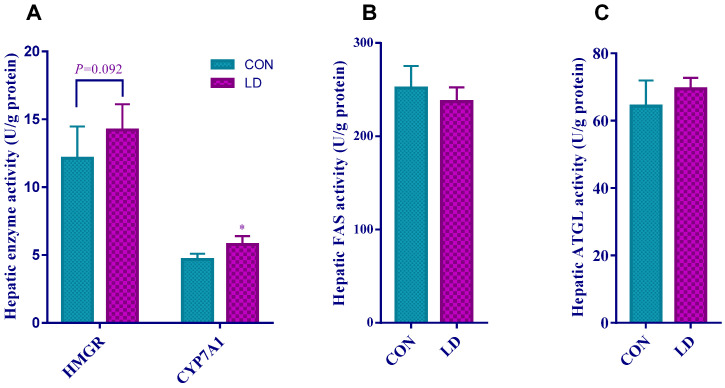
*L. delbrueckii* affected hepatic enzyme activity related to cholesterol and bile acid metabolism. Hepatic activities of HMGR and CYP7A1 (**A**), FAS (**B**), and ATGL (**C**) in Ningxiang pigs. Values in columns are shown as the mean ± SD, * *p* < 0.05. HMGR = β-hydroxy-β-methylglutarate monoyl coenzyme A reductase; CYP7A1 = cholesterol-7α hydroxylase; FAS = fatty acid synthetase; ATGL = adipose triglyceride lipase; CON = basal diets; LD = basal diets + 0.1% *L. delbrueckii*.

**Figure 4 animals-14-01801-f004:**
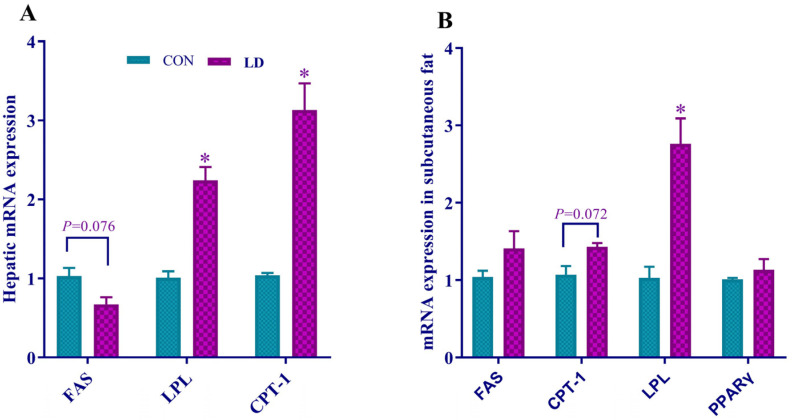
*L. delbrueckii* regulates gene expression related to fatty acid synthesis, lipolysis, and *β*-oxidation in selected tissues. Gene expression is related to fat metabolism in the liver (**A**) and subcutaneous fat (**B**) in Ningxiang pigs. Values in columns are shown as the mean ± SD, * *p* < 0.05. *FAS* = fatty acid synthetase; *LPL* = lipoprotein lipase; *CPT-1* = carnitine palmitoyltransferase; *PPARγ* = peroxidase proliferative receptor γ; CON = basal diets; LD = basal diets + 0.1% *L. delbrueckii*.

**Figure 5 animals-14-01801-f005:**
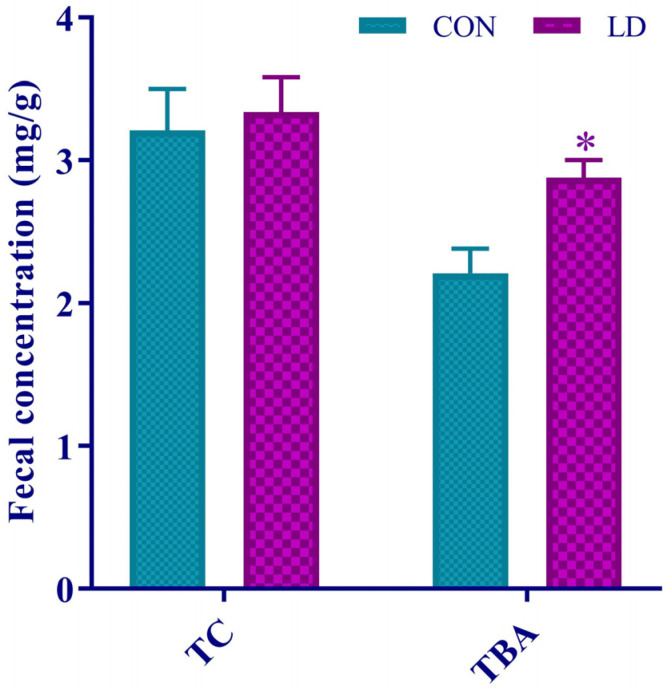
*L. delbrueckii* facilitated fecal TBA excretion. Fecal concentrations of TC and TBA in Ningxiang pigs. Values in columns are shown as the mean ± SD, * *p* < 0.05. TC = total cholesterol; TBA = total bile acid; CON = basal diets; LD = basal diets + 0.1% *L. delbrueckii*.

**Figure 6 animals-14-01801-f006:**
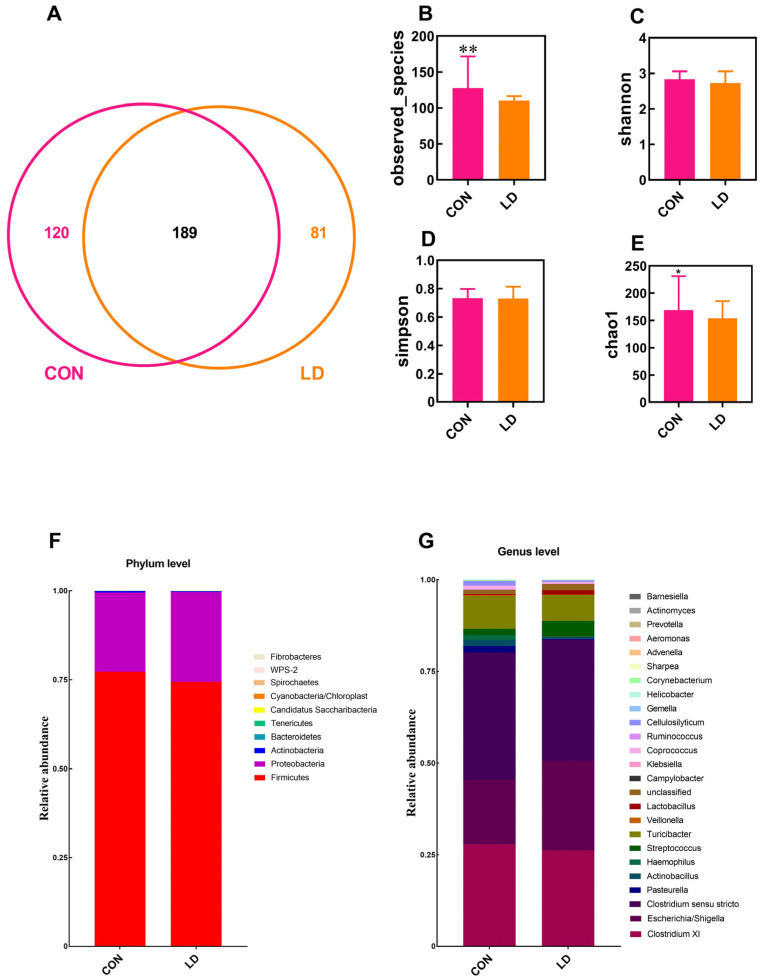
Ileal microbiota composition. Venn picture (**A**), α-diversity (**B**–**E**), and relative abundance at the phylum (**F**) and genus (**G**) levels of ileal microbiota in Ningxiang pigs. Values in columns are shown as the mean ± SD, * *p* < 0.05, ** *p* < 0.01. CON = basal diets; LD = basal diets + 0.1% *L. delbrueckii*.

**Figure 7 animals-14-01801-f007:**
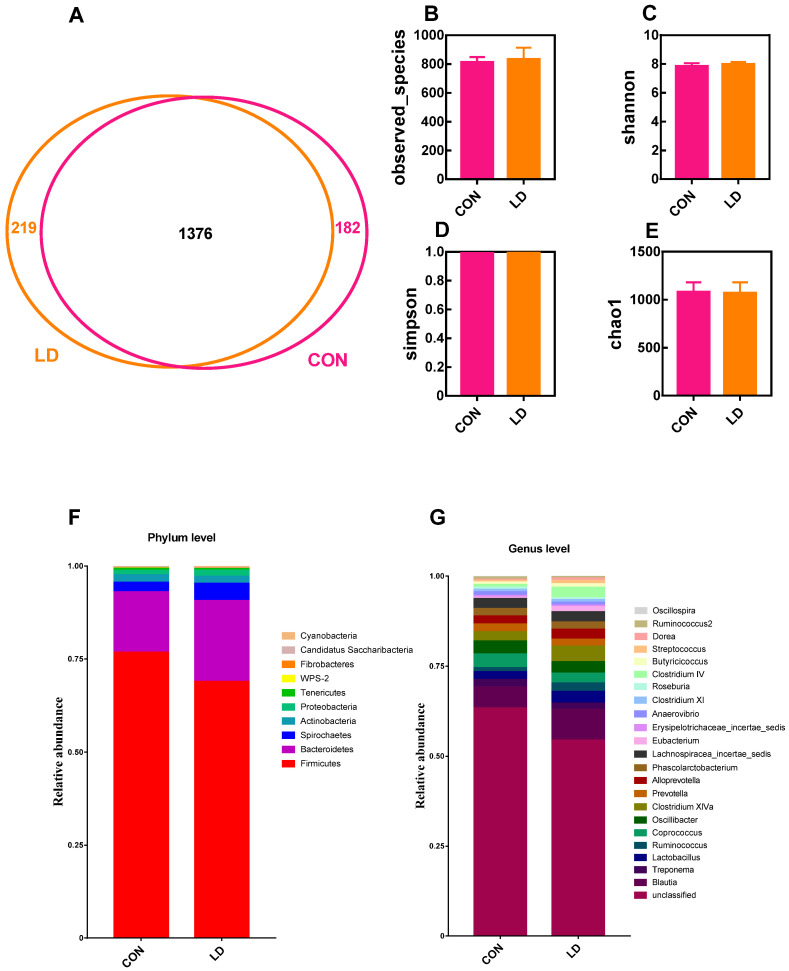
Colonic microbiota composition. Venn picture (**A**), α-diversity (**B**–**E**), and relative abundance at the phylum (**F**) and genus (**G**) levels of the colonic microbiota in Ningxiang pigs. Values in columns are shown as the mean ± SD, CON = basal diets; LD = basal diets + 0.1% *L. delbrueckii*.

**Figure 8 animals-14-01801-f008:**
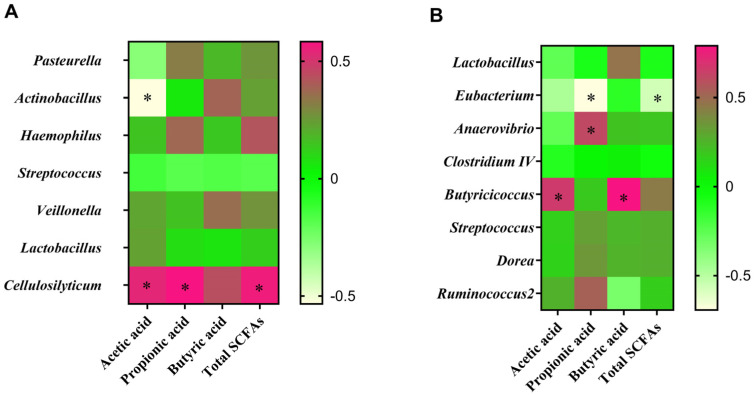
Correlation between gut microbiota composition and SCFA profiles. Pearson correlation analysis between the differential microbiota and measured SCFAs in the ileum (**A**) and colon (**B**). Red represents a positive correlation, while light yellow indicates a negative correlation. The asterisk (*) shows a significant difference (*p* < 0.05). CON = basal diets; LD = basal diets + 0.1% *L. delbrueckii*.

**Table 1 animals-14-01801-t001:** Ingredient composition and nutrient levels of the basal diet (air-dry basis, %).

Items	30–50 kg	50–80 kg
Ingredients		
Corn	51.30	45.50
Soybean meal (CP, 43%)	12.70	8.20
Paddy rice	15.70	23.50
Rice bran meal	18.00	20.50
Limestone	0.70	0.70
CaHPO_4_	0.80	0.80
NaCl	0.30	0.30
Premix ^1^	0.50	0.50
Total	100.00	100.00
Calculated nutritional levels		
Digestible energy, DE MJ/kg	13.07	12.64
Crude protein, CP	13.51	12.01
Standardized ileal digestible lysine, SID Lys	0.53	0.43
Standardized ileal digestible methionine, SID Met	0.22	0.20
Standardized ileal digestible threonine, SID Thr	0.40	0.34
Standardized ileal digestible tryptophan, SID Trp	0.11	0.09
Calcium, Ca	0.57	0.56
Available phosphorus, AP	0.42	0.44
Analyzed nutritional levels		
Gross energy, GE MJ/kg	14.87	14.31
Crude protein, CP	13.72	12.27
Ether extract, EE	2.42	2.35

^1^ The premix provided the following per kg of diet: vitamin A, 4 060 IU; vitamin D_3_, 2 030 IU; vitamin E, 40.6 IU; vitamin K_3_, 4.0 mg; vitamin B_12_, 20.0 μg; vitamin B_1_, 2.0 mg; vitamin B_2_, 8.0 mg; pantothenic acid, 15.2 mg; niacin, 20.3 mg; choline chloride, 609 mg; Mn (manganese oxide), 61.8 mg; Fe (ferrous sulfate), 109.2 mg; Zn (zinc oxide), 104 mg; Cu (copper sulfate), 22 mg; I (potassium iodide) 0.3 mg; Se (sodium selenite), 0.31 mg.

**Table 2 animals-14-01801-t002:** Primers for Target Genes.

Gene	Accession Number	Sequences (5′-3′)	Product Size (pb)
*CPT-1*	NM_001129805.1	F: GACAAGTCCTTCACCCTCATCGC	117
		R: GGGTTTGGTTTGCCCAGACAG	
*PPARγ*	NM_214379.1	F: ACTTTATGGAGCCCAAGTTCG	108
		R: GCAGCAAATTGTCTTGAATGTCC	
*LPL*	NM_214286.1	F: CACATTCACCAGAGGGTC	126
		R: TCATGGGAGCACTTCACG	
*FAS*	NM_213839.1	F: TTTTCCCTGGCACTGGCTACCTG	81
		R: TGCAGCGTCACGTCCTCAAACAC	
*GAPDH*	NM_001206359.1	F: ATGGTGAAGGTCGGAGTGAAC	235
		R: CTCGCTCCTGGAAGATGGT	

*CPT-1* = Carnitine palmitoyltransferase; *PPARγ* = Peroxidase proliferative receptor γ; *LPL* = Lipoprotein lipase; *FAS* = Fatty acid synthetase; *GAPDH* = glyceraldehyde-3-phosphate dehydrogenase.

**Table 3 animals-14-01801-t003:** Short-chain fatty acid (SCFA) profiles in the gut contents of Ningxiang pigs (mg/g).

Items	CON	LD	*p*-Value
Ileal digesta			
Acetic acid	0.671 ± 0.165 ^a^	0.374 ± 0.198 ^b^	0.018
Propionic acid	0.039 ± 0.029	0.060 ± 0.043	0.329
Butyric acid	0.067 ± 0.060	0.074 ± 0.057	0.837
Total SCFAs	0.775 ± 0.232	0.506 ± 0.200	0.058
Colonic digesta			
Acetic acid	0.509 ± 0.177	0.680 ± 0.149	0.101
Propionic acid	0.532 ± 0.098	0.319 ± 0.250	0.080
Butyric acid	0.239 ± 0.060 ^a^	0.597 ± 0.238 ^b^	0.045
Total SCFAs	1.280 ± 0.259	1.610 ± 0.581	0.063

CON = basal diets; LD = basal diets + 0.1% *L. delbrueckii*. Letters a and b in the rows describe significant differences between treatments at *p* < 0.05.

## Data Availability

The data presented in this study are available in this article.

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
