# Peer review of "Lactobacillus delbrueckii* Ameliorated Blood Lipids via Intestinal Microbiota Modulation and Fecal Bile Acid Excretion in a Ningxiang Pig Model"

_animals, 2024, doi:10.3390/ani14121801_

Round 1

Reviewer 1 Report

Comments and Suggestions for Authors

Lactobacillus Delbrueckii Ameliorated Blood Lipids via Intestinal Microbiota Modulation and Fecal Bile Acid Excretion in a Ninxiang Pig Model

Dear Authors,

Manuscript is very interesting and well prepared, but there are some corrections required.

Below I add some suggestions helpful in this process:

Line 15

Lack of Simple summary.

Line 83

Table 1

In case of soybean meal content of crude protein is important: Soybean meal (CP ?%)

In case of last ingredient: Pavailable is normally used in case of description

Line 157

Information about Shapiro-Wilk’s test (normal distribution of data in treatment) must be also added.

Line 171, 181, 192, 204, 212, 229, 246

Histograms word must be changed to columns. Because histograms present frequency distributions of data in sample gathered from population (one treatment in experiment) and eventually compare it with normal distribution curve, using bins and frequences in each bin of histogram.

Ie: values in columns are presented as a mean±SD.

Line 254

Significance level could be emphasized in case of acetic acid in illeal digesta and butyric acid in colonic digesta:

Acetic acid

0.671±0.165a

0.374±0.198b

0.018

Butyric acid

0.239±0.060b

0.597±0.238a

0.045

CON… L.delbrueckii. Letters a,b in rows describe significant differences between treatments at p<0.05  (or in a form of asterisk similar as of figures with significance description)

Line 285

Space after ‘industry’ word required, and form [23-25] (three in a row references or more, according with Instructions for Authors) instead of [23, 24, 25].

Line 347

Conclusions must be present in form of sentences instead of graphical form. Maybe it is possible to use this graphic on the beginning of article with standard abstract, as an graphical abstract.

Line 376-471

Doi number/link required on the end of reference.

Line 389

Abbreviation of Journal required: Adv. Drug Deliv Rev.

Line 395

Number of “page” required: …, 2020, 25, 5656

Line 398

Number of “page” required: …, 2022, 9, 982349

Line 404

In text is: J. App.l Toxicol, must be J. Appl. Toxicol.

Line 406

Number of “page” required: …, 2020, 15, e0236629

Line 428

Second and third row of reference must be in the same line as first.

Line 462

Number of “page” required: …, 2022, 14, 4620

Line 463 and 466

Same like in line 428.

Author Response

# reviewer 1

Dear Authors,

Manuscript is very interesting and well prepared, but there are some corrections required.

Below I add some suggestions helpful in this process:

Thank you very much for your careful comments and valuable suggestions. All of the comments, questions and opinions were considered carefully, and the manuscript has been revised accordingly.

Line 15

Lack of Simple summary.

Response: thank you very much for your careful comments and valuable suggestion. We have added it in the revised manuscript.

Line 83

Table 1

In case of soybean meal content of crude protein is important: Soybean meal (CP ?%)

Response: thank you very much for your careful comments and valuable suggestion. We have added the information in the Table 1.

In case of last ingredient: Pavailable is normally used in case of description

Response: thank you very much for your careful comments and valuable suggestion. We know that available phosphorus (AP) is also widely used in the NRC (2012) or many published articles. If the Journal (Animals) requires a uniform expression format, we can modify it.

Line 157

Information about Shapiro-Wilk’s test (normal distribution of data in treatment) must be also added.

Response: thank you very much for your careful comments and valuable suggestion. We have added it in the section of Statistics Analysis.

Line 171, 181, 192, 204, 212, 229, 246

Histograms word must be changed to columns. Because histograms present frequency distributions of data in sample gathered from population (one treatment in experiment) and eventually compare it with normal distribution curve, using bins and frequences in each bin of histogram.

Ie: values in columns are presented as a mean±SD.

Response: thank you very much for your careful comments and valuable suggestion. We have changed them in the revised manuscript.

Line 254

Significance level could be emphasized in case of acetic acid in illeal digesta and butyric acid in colonic digesta:

Acetic acid

0.671±0.165a

0.374±0.198b

0.018

Butyric acid

0.239±0.060b

0.597±0.238a

0.045

CON… L.delbrueckii. Letters a,b in rows describe significant differences between treatments at p<0.05  (or in a form of asterisk similar as of figures with significance description)

Response: thank you very much for your careful comments and valuable suggestion. We have changed them in the revised manuscript.

Line 285

Space after ‘industry’ word required, and form [23-25] (three in a row references or more, according with Instructions for Authors) instead of [23, 24, 25].

Response: thank you very much for your careful comments and valuable suggestion. We have revised them in the text.

Line 347

Conclusions must be present in form of sentences instead of graphical form. Maybe it is possible to use this graphic on the beginning of article with standard abstract, as an graphical abstract.

Response: thank you very much for your careful comments and valuable suggestion. We have deleted the picture and put it in the supplementary materials.

Line 376-471

Doi number/link required on the end of reference.

Response: thank you very much for your careful comments and valuable suggestion. We have added the doi number for all references except the reference ‘Li, D. F.; Wang, K.L.; Qiao, S. Y.; Jia, G.; Jiang, Z. Y.; Chen, Z. L.; Lin, Y. C.; Wu, D.; Zhu, X. M.; Xiong, B. H.; Yang, L. B.; Wang, F. L. Feeding standard of swine, the agricultural industry standard of the People’s Republic of China (NY/T 65-2004), 2004. (in Chinese)’, which is a standard lacking of the number.

Line 389

Abbreviation of Journal required: Adv. Drug Deliv Rev.

Response: thank you very much for your careful comments and valuable suggestion. We have revised it.

Line 395

Number of “page” required: …, 2020, 25, 5656

Response: thank you very much for your careful comments and valuable suggestion. We have added it.

Line 398

Number of “page” required: …, 2022, 9, 982349

Response: thank you very much for your careful comments and valuable suggestion. We have added it.

Line 404

In text is: J. App.l Toxicol, must be J. Appl. Toxicol.

Response: thank you very much for your careful comments and valuable suggestion. We have added it.

Line 406

Number of “page” required: …, 2020, 15, e0236629

Response: thank you very much for your careful comments and valuable suggestion. We have added it.

Line 428

Second and third row of reference must be in the same line as first.

Response: thank you very much for your careful comments and valuable suggestion. We have revised it.

Line 462

Number of “page” required: …, 2022, 14, 4620

Response: thank you very much for your careful comments and valuable suggestion. We have added it.

Line 463 and 466

Same like in line 428.

Response: thank you very much for your careful comments and valuable suggestion. We have revised it.

Reviewer 2 Report

Comments and Suggestions for Authors

This manuscript investigated the effect of Lactobacillus delbrueckii addition on blood lipids in Ningxiang pigs.  My comments are as follows:

Line 43-50: This study is not about researching the nutritional value or function of pork, and writing about that is disconnected from the topic of the article. The authors need to describe the mechanisms and factors influencing the formation of hyperlipidemia. What are the means of regulating hyperlipidemia in the diet?

Line 51-55: First, the purpose of the study was unclear. Was this study trying to address the animal husbandry problem of lean body mass or intramuscular fat or the medical problem of high blood lipids? If it is a study of medical problems, the Ningxiang pig is only used as a model of a fatty pig breed, and there is no need to introduce so much about Chinese local pig breeds. Just to introduce the fact that Ningxiang pigs are fat pigs, what is the backfat thickness of an adult pig? What is the lean meat percentage?

Line 59: The authors have previously done similar studies on lean pigs and published three papers. The authors should present: what is different about this study on fatty pigs? Is it just a different pig breed? What are the differences in fat metabolism and mechanisms of hyperlipidemia formation between lean and fat pigs? What are the different results that can only be obtained by experimenting with fatty pigs?

Line 70: The age of these pigs?

Line 75: When did these pigs reach 50kg body weight? If the authors did not weigh them, how do they know when to change feed? If these pigs were weighed, why didn't the authors show their growth performance data?

Line 72-73: How is L. delbrueckii added to pig feed? How is it mixed? Is the feed pelleted? How much of this L. delbrueckii product can be stored at room temperature? Are they coated to prevent the negative effects of stomach acid? Has the product been tested in vitro in a perigastric test?

2.2. Animals, Diets and Experimental Design.

The authors concluded that feeding Lactobacillus delbrueckii reduced hepatic and serum lipid, so why did the authors not record the backfat thickness of these pigs? If hepatic and serum lipid decreased, would this result in less fat deposition in the pigs?

The authors also did not measure the intramuscular fat content of the pigs. The key results throughout the manuscript are only the results of Elisa kits. In fact, Elisa results are very variable depending on product quality. These bar graph results are not convincing enough and it would have been better if the authors could have provided some immunofluorescence, stained sections or WB plots.

The discrepancy in the 16s RNA gene sequencing results was entirely predictable because the authors fed Lactobacillus delbrueckii that would surely have had some effects on the GI microbes. Adding a type of microorganism that makes a change in microbial composition is not considered a key result or its molecular mechanism.

In addition, during the 60-day trial period, why did the authors not record the growth traits of these pigs, including weight gain and feed intake, which could have been calculated for feed conversion efficiency. If more feed energy was not used to synthesize fat, it could have been used to synthesize protein, which might have changed the feed conversion ratio.

In the gut microbiology results, I'd like to see what the results were for each of the 12 pigs in the two groups. How consistent was the gut microbial composition of the 6 pigs in the group?

Author Response

# reviewer 2

This manuscript investigated the effect of Lactobacillus delbrueckii addition on blood lipids in Ningxiang pigs. My comments are as follows:

 Thank you very much for your careful comments and valuable suggestions. All of the comments, questions and opinions were considered carefully, and the manuscript has been revised accordingly.

Line 43-50: This study is not about researching the nutritional value or function of pork, and writing about that is disconnected from the topic of the article. The authors need to describe the mechanisms and factors influencing the formation of hyperlipidemia. What are the means of regulating hyperlipidemia in the diet?

 Thank you very much for your careful comments and valuable suggestions. We have carefully checked and revised them in the manuscript.

Line 51-55: First, the purpose of the study was unclear. Was this study trying to address the animal husbandry problem of lean body mass or intramuscular fat or the medical problem of high blood lipids? If it is a study of medical problems, the Ningxiang pig is only used as a model of a fatty pig breed, and there is no need to introduce so much about Chinese local pig breeds. Just to introduce the fact that Ningxiang pigs are fat pigs, what is the backfat thickness of an adult pig? What is the lean meat percentage?

 Thank you very much for your careful comments and valuable suggestions. Our objective mainly focus on the medical problem of high blood lipids, and we have carefully checked and revised them in the manuscript.

Line 59: The authors have previously done similar studies on lean pigs and published three papers. The authors should present: what is different about this study on fatty pigs? Is it just a different pig breed? What are the differences in fat metabolism and mechanisms of hyperlipidemia formation between lean and fat pigs? What are the different results that can only be obtained by experimenting with fatty pigs?

 Thank you very much for your careful comments and valuable suggestions. We have compared the differences between Niangxiang pigs and lean pigs, and found that the serum lipids levels (TG and TC) are higher in Ningxiang pigs, which might ascribe to their stronger fat deposition capacity and higher fat rate (about 46% in adult pigs, VS 17.8 %) and backfat thickness (about 51 mm, VS 20mm) (Hou et al., 2016; Wei et al., 2017). Therefore, Ningxiang pigs, a typical fatty pig breed, might be more suitable as a model to study hyperlipidemia compared with lean pigs

Hou G F, Li R, Liu M, et al.Effects of Lactobacillus delbrueckii on Carcass Traits and Meat Quality of Fattening Pigs. Chinese J. Anim. Nutr. 2016, 28, 1814-1822. doi: 10.3969 /j.issn.1006-267x.2016.06.024.(in Chinese)

Wei,L.; Li,R.; Liu, M.; Wang, H.; Hou, S.; Hou, G.; Huang,X. Effects of lactobacillus delbrueckii on performance, carcass traits and meat quality of Ningxiang pigs. Chinese J. Anim. Nutr. 2017, 29, 4562-4569. doi: 10.3969 /j.issn.1006-267x.2017.12.038.(in Chinese)

Line 70: The age of these pigs?

 Thank you very much for your careful comments and valuable suggestions. The age of these pigs was 110 ± 2 days, and these information was added in the revised manuscript.

Line 75: When did these pigs reach 50kg body weight? If the authors did not weigh them, how do they know when to change feed? If these pigs were weighed, why didn't the authors show their growth performance data?

 Thank you very much for your careful comments and valuable suggestions. Our group have completed a lot of research works in Ningxiang pigs, and one of researches focuses on their growth changes with age based on the automatic feeding system (see the table 1 and figure 1 below) (Zhou et al., 2024). Therefore we know when the pigs reached about 50 kg BW and change the feed after we conducted the experiment. Additionally, we also recorded the weight of Ningxiang pigs (see the table 2 below), however, these data have been used in another paper, which has been submitted to the Chinese Journal of Animal Nutrition (Wei et al., 2024). In order to avoid the duplicate submission, we did not put these data in the manuscript.

Table 1 Ningxiang pig growth with their age

Figure 1 The automatic feeding system for nutritional research in Niangxiang pigs

Zhou X, Guo H M, Yang F,et al. Growth and development law and growth curve fitting analysis of Ningxiang pigs. Chinese Journal of Animal Science, 2024.(Accepted)

Wei L, Hou G F Huang X G, et al. Effects of lactobacillus delbrueckii on performance, carcass traits and meat quality of Ningxiang pigs. Chinese Journal of Animal Nutrition, 2024.(Accepted)

Line 72-73: How is L. delbrueckii added to pig feed? How is it mixed? Is the feed pelleted? How much of this L. delbrueckii product can be stored at room temperature? Are they coated to prevent the negative effects of stomach acid? Has the product been tested in vitro in a perigastric test?

 Thank you very much for your careful comments and valuable suggestions. The strain was premixed with corn meal via stepwise dilution to reach the condition of feed mixing. Mash feed was provided during the experimental period. A pack of L. delbrueckii product is 1kg and the strain was coated with sodium alga acid. A total of 50kg of products were produced by the PERFLY-BIO (Changsha, China) and stored in -20℃refrigeration house before feed preparation. We usually produce one batch of feed every two weeks. We have done a lot of in vitro biological characteristics of the strains (Figure 1-4 and table 1). Additionally, we have conducted a series of researches on L. delbrueckii using in pig production and most of papers have been published (you can get some information in the web).

Figure 1. Colony morphological characteristics of lactobacillus delbrueckii

Figure 2. Growth curve of Lactobacillus delbrueckii

Figure 3. Acid-base resistance of Lactobacillus delbrueckii

Figure 4. Indentification of bile salt hydrolase (BSH) activity in Lactobacillus delbrueckii. Incubation on MRS plate containing bile salts for 12 h (A) and 72 h (B), respectively. Amplification of BSH genes in Lactobacillus delbrueckii (C), target band sites for BSH1 and BSH2 gene were markered with blue and yellow box, respectively.

Figure 5. Cholesterol clearance of Lactobacillus delbrueckii in vitro. TC contents in the culture before and after the treatment (A);Cholesterol clearance (B), “*”represented significantly differences (P<0.05); “**” differed extremely (P<0.05),respectively.

Table 2-2 Number of Lactobacillus in ileal and colonic mucosa(log10 of cells/g of sample)

Items

control group

experimental group

P-Value

Ileal mucosa

5.7±0.13

6.3±0.27*

0.047

Colonic mucosa

7.3±0.09

7.8±0.49*

0.036

2.2. Animals, Diets and Experimental Design.

The authors concluded that feeding Lactobacillus delbrueckii reduced hepatic and serum lipid, so why did the authors not record the backfat thickness of these pigs? If hepatic and serum lipid decreased, would this result in less fat deposition in the pigs?

 Thank you very much for your careful comments and valuable suggestions. We measured the carcass trait of Ningxiang pigs (see the table 3 below), however, these data have been used in another paper, which has been submitted to the Chinese Journal of Animal Nutrition (Wei et al., 2024). In order to avoid the duplicate submission, we did not put these data in the manuscript. Interestingly, we indeed found the fat rate and backfat thickness were numerically reduced by Lactobacillus delbrueckii, which echoed the hepatic and serum results.

Wei L, Hou G F Huang X G, et al. Effects of lactobacillus delbrueckii on performance, carcass traits and meat quality of Ningxiang pigs. Chinese Journal of Animal Nutrition, 2024.(Accepted)

The authors also did not measure the intramuscular fat content of the pigs. The key results throughout the manuscript are only the results of Elisa kits. In fact, Elisa results are very variable depending on product quality. These bar graph results are not convincing enough and it would have been better if the authors could have provided some immunofluorescence, stained sections or WB plots.

 Thank you very much for your careful comments and valuable suggestions. We determined the IMF (see table 5 below), however, these data have been used in another paper, which has been submitted to the Chinese Journal of Animal Nutrition (Wei et al., 2024). In order to avoid the duplicate submission, we did not put these data in the manuscript. Additionally, we plan to do some analysis according to the expert advice, if the funds are sufficient.

Wei L, Hou G F Huang X G, et al. Effects of lactobacillus delbrueckii on performance, carcass traits and meat quality of Ningxiang pigs. Chinese Journal of Animal Nutrition, 2024.(Accepted)

The discrepancy in the 16s RNA gene sequencing results was entirely predictable because the authors fed Lactobacillus delbrueckii that would surely have had some effects on the GI microbes. Adding a type of microorganism that makes a change in microbial composition is not considered a key result or its molecular mechanism.

 Thank you very much for your careful comments and valuable suggestions. Indeed, we need further to conduct experiments to explore the mechanism and we plan to use the rat model to validate the results and reveal the mechanism.

In addition, during the 60-day trial period, why did the authors not record the growth traits of these pigs, including weight gain and feed intake, which could have been calculated for feed conversion efficiency. If more feed energy was not used to synthesize fat, it could have been used to synthesize protein, which might have changed the feed conversion ratio.

 Thank you very much for your careful comments and valuable suggestions.

In the gut microbiology results, I'd like to see what the results were for each of the 12 pigs in the two groups. How consistent was the gut microbial composition of the 6 pigs in the group?

Thank you very much for your careful comments and valuable suggestions. We provide microbiota composition in ileal or colonic digesta of all selected pigs (figure 1 and 2), however, the results from the picture are confusing due to great differences among 12 individual pig. To simplify the results, we combined the same set of data and presented it as Mean±SD (table S1 and S2). Therefore, we use the original figures better.

Figure 1. Ileal microbiota composition.

Figure 2. Colonic microbiota composition.

Table S1. Microbiota Composition at Phylum and Genus Level in Ileal Digesta

Items

CON

LD

P value

Phylum level

Firmicutes

0.8113±0.1783

0.7452±0.2119

0.706

Proteobacteria

0.1839±0.1792

0.2514±0.2108

0.714

Actinobacteria

0.0042±0.0033

0.0020±0.0001

0.426

Bacteroidetes

0.0002±0.0002

0

0.051

Tenericutes

0.0001±0.0001

0.0002±0.0003

0.163

Candidatus Saccharibacteria

0.0002±0.0004

0.0011±0.0014

0.129

Cyanobacteria/Chloroplast

0.0001±0.0001

0

0.280

Spirochaetes

0

0

0.024

WPS-2

0

0

Fibrobacteres

0

0

Genus level

Clostridium XI

0.2532±0.1523

0.2682±0.1417

0.846

Escherichia/Shigella

0.1460±0.1267

0.2439±0.2094

0.320

Clostridium sensu stricto

0.4024±0.2549

0.3272±0.1894

0.483

Pasteurella

0.0143±0.0269

0.0020±0.0021

0.033

Actinobacillus

0.0134±0.0225

0.0055±0.0037

0.050

Haemophilus

0.0090±0.0157

0.0010±0.0018

0.041

Streptococcus

0.0164±0.0073

0.0408±0.0405

0.001

Turicibacter

0.0935±0.0583

0.0702±0.0626

0.600

Veillonella

0.0029±0.0028

0.0012±0.0011

0.065

Lactobacillus

0.0026±0.0014

0.0114±0.0170

0.038

unclassified

0.0140±0.0116

0.0172±0.0119

0.787

Campylobacter

0.0002±0.0004

0

0.032

Klebsiella

0.0007±0.0005

0.0008±0.0008

0.094

Coprococcus

0.0115±0.0127

0.0045±0.0039

0.166

Ruminococcus

0.0005±0.0003

0.0001±0.0001

0.298

Cellulosilyticum

0.0158±0.0143

0.0044±0.0025

0.004

Gemella

0.0005±0.0002

0.0006±0.0007

0.164

Helicobacter

0.0001±0.0001

0

0.003

Corynebacterium

0.0024±0.0018

0.0009±0.0005

0.081

Sharpea

0.0001±0.0001

0.0001±0.0001

0.930

Advenella

0.0001±0.0001

0.0001±0

0.054

Aeromonas

0

0

0.024

Prevotella

0.0001±0.0001

0

0.000

Actinomyces

0.0002±0.0002

0.0003±0.0004

0.347

Barnesiella

0

0

0.002

Table S2. Microbiota Composition at Phylum and Genus Level in Colonic Digesta

Items

CON

LD

P value

Phylum level

Firmicutes

0.7634±0.0654

0.6853±0.0776

0.743

Bacteroidetes

0.1667±0.0677

0.2222±0.0591

0.936

Spirochaetes

0.0261±0.0166

0.0473±0.0207

0.512

Actinobacteria

0.0196±0.0053

0.0180±0.0066

0.348

Proteobacteria

0.0119±0.0119

0.0163±0.0080

0.128

Tenericutes

0.0055±0.0026

0.0038±0.0012

0.010

WPS-2

0.0010±0.0010

0.0003±0.0003

0.001

Fibrobacteres

0.0006±0.0003

0.0008±0.0005

0.221

Candidatus Saccharibacteria

0.0019±0.0021

0.0006±0.0005

0.069

Cyanobacteria

0.0035±0.0029

0.0055±0.0038

0.306

Genus level

0.7634±0.0654

0.6853±0.0776

0.743

unclassified

0.6288±0.0647

0.5431±0.0379

0.395

Blautia

0.0608±0.0410

0.0884±0.0354

0.771

Treponema

0.0213±0.1270

0.0173±0.0134

0.952

Lactobacillus

0.0216±0.0133

0.0311±0.0191

0.065

Ruminococcus

0.0123±0.0164

0.0241±0.0127

0.658

Coprococcus

0.0374±0.0080

0.0266±0.0083

0.740

Oscillibacter

0.0363±0.0130

0.0317±0.0078

0.004

Clostridium XlVa

0.0264±0.0106

0.0429±0.0100

0.964

Prevotella

0.0216±0.0045

0.0186±0.0043

0.736

Alloprevotella

0.0224±0.0072

0.0286±0.0076

0.875

Phascolarctobacterium

0.0205±0.0054

0.0194±0.0078

0.156

Lachnospiracea_incertae_sedis

0.0273±0.0149

0.0279±0.0149

0.824

Eubacterium

0.0042±0.0036

0.0152±0.0154

0.096

Erysipelotrichaceae_incertae_sedis

0.0050±0.0031

0.0042±0.0033

0.620

Anaerovibrio

0.0098±0.0033

0.0084±0.0011

0.001

Clostridium XI

0.0071±0.0034

0.0082±0.0041

0.672

Roseburia

0.0057±0.0044

0.0039±0.0023

0.194

Clostridium IV

0.0076±0.0032

0.0303±0.0416

0.056

Butyricicoccus

0.0070±0.0004

0.0088±0.0076

0.032

Streptococcus

0.0044±0.0025

0.0095±0.0109

0.073

Dorea

0.0030±0.0023

0.0062±0.0085

0.087

Ruminococcus2

0.0069±0.0030

0.0044±0.0006

0.065

Oscillospira

0.0028±0.0022

0.0014±0.0024

0.732

Reviewer 3 Report

Comments and Suggestions for Authors

General comment

The manuscript is original, with information relevant to science. The figures presenting the results are very well assembled, with excellent resolution; But I have one criticism, which is the very small size of the asterisk (*) that identifies the differences. Adjust this.

Larger comments:

1) the summary section was very succinct, with a sentence presenting results, without a brief description of M&M. Furthermore, the authors speak of a microbiota focus; but they do not present any results in this section, which allow us to understand the modulation carried out by Lactobacillus that has the capacity to alter blood lipids.

2) In the introduction section "The lipid-lowering effects of Lactobacillus delbrueckii in lean pigs have been confirmed in our previous studies [3, 9, 15], however, the role of the strain in lard type pigs is unclear." I want to invite the authors to develop this idea further, because the hypothesis for this research should emerge from it.

3) In the M&M section, the authors say "Pigs in each pen were weighed together at the start and end of each period, and the daily feed consumption per pen was recorded." However, these data were not presented in the results; and are completely necessary in this research; We need to understand what happened to the development of animals. In addition to weight and feed intake, authors must calculate feed conversion and present this data.

4) The authors presented the nutritional composition of the diet in a calculated way; but the ideal would be to present the analysis. The ingredients used to produce the diet are almost always very different from those used in the formulation. As your work focuses directly on nutrition and the intestine; This should be done and also present analyzed data on crude protein, crude ebergia and mainly ethereal extract.

5) in the statistical analysis, which is very timid, it is not clear how the microbiota analyzes were carried out; the authors need to make this very detailed; the tests carried out, which database was used to determine relative abundance, among other variables.

6) results and discussion section were good; has clear information.

7) the conclusion section was very different from anything I've ever seen; it is presented in the form of a diagram (almost a graphic abstract). In my opinion, this should be the last paragraph of the discussion section; there in the conclusion section, only in text form; authors must “answer the objectives”.

Author Response

# reviewer 3

The manuscript is original, with information relevant to science. The figures presenting the results are very well assembled, with excellent resolution; But I have one criticism, which is the very small size of the asterisk (*) that identifies the differences. Adjust this.

 Thank you very much for your careful comments and valuable suggestions. We have checked carefully and revised them throughout the manuscript.

Larger comments:

1) the summary section was very succinct, with a sentence presenting results, without a brief description of M&M. Furthermore, the authors speak of a microbiota focus; but they do not present any results in this section, which allow us to understand the modulation carried out by Lactobacillus that has the capacity to alter blood lipids.

Thank you very much for your careful comments and valuable suggestions. We have partly revised the abstract according to the comments, however, we mostly stuck with the original idea, which was more succinct. Because most of description are shown in the results and discussion section, and it unnecessary to put them again in the abstract.

2) In the introduction section "The lipid-lowering effects of Lactobacillus delbrueckii in lean pigs have been confirmed in our previous studies [3, 9, 15], however, the role of the strain in lard type pigs is unclear." I want to invite the authors to develop this idea further, because the hypothesis for this research should emerge from it.

Thank you very much for your careful comments and valuable suggestions. We will conduct some rat experiments to further explore the mechanism.

3) In the M&M section, the authors say "Pigs in each pen were weighed together at the start and end of each period, and the daily feed consumption per pen was recorded." However, these data were not presented in the results; and are completely necessary in this research; We need to understand what happened to the development of animals. In addition to weight and feed intake, authors must calculate feed conversion and present this data.

Thank you very much for your careful comments and valuable suggestions. We did the work and recorded the weight of Ningxiang pigs (see the Table 2 below), however, these data have been used in another paper, which has been submitted to the Chinese Journal of Animal Nutrition (Wei et al., 2024). In order to avoid the duplicate submission, we did not put these data in the manuscript.

Wei L, Hou G F Huang X G, et al. Effects of lactobacillus delbrueckii on performance, carcass traits and meat quality of Ningxiang pigs. Chinese Journal of Animal Nutrition, 2024.(Accepted)

4) The authors presented the nutritional composition of the diet in a calculated way; but the ideal would be to present the analysis. The ingredients used to produce the diet are almost always very different from those used in the formulation. As your work focuses directly on nutrition and the intestine; This should be done and also present analyzed data on crude protein, crude ebergia and mainly ethereal extract.

Thank you very much for your careful comments and valuable suggestions. In the last 2 weeks, we completed the analysis work of approximate nutrients in the diets and put these analyzed value in the Table 1 in the revised manuscript.

5) in the statistical analysis, which is very timid, it is not clear how the microbiota analyzes were carried out; the authors need to make this very detailed; the tests carried out, which database was used to determine relative abundance, among other variables.

Thank you very much for your careful comments and valuable suggestions. We have changed and added them in the revised manuscript.

6) results and discussion section were good; has clear information.

Thank you very much for your careful comments and valuable suggestions.

7) the conclusion section was very different from anything I've ever seen; it is presented in the form of a diagram (almost a graphic abstract). In my opinion, this should be the last paragraph of the discussion section; there in the conclusion section, only in text form; authors must “answer the objectives”.

Thank you very much for your careful comments and valuable suggestions. We have deleted the picture and put it in the supplementary materials.

Round 2

Reviewer 2 Report

Comments and Suggestions for Authors

The authors have addressed my questions before. 

Author Response

thank you for your valuable suggestion.

Reviewer 3 Report

Comments and Suggestions for Authors

Adjustments made. One doubt that remained is regarding this experiment having given rise to two articles, and in the response letter it was informed that "Wei L, Hou G F Huang X G, et al. Effects of lactobacillus delbrueckii on performance, carcass traits and meat quality of Ningxiang pigs. Chinese Journal of Animal Nutrition, 2024.(Accepted)"

in the reference section of the article (ref. 28) there is a reference with the same authors and title, however with the year of publication 2017 (I found it online).

It needs to be very clear in the methodology section that these results refer to an experiment carried out and previously published (cite the work in M&M).

Information about body weight or weight gain must be in this manuscript, it can be descriptive and referencing the article by Wei et al.

Adjust

Author Response

thank your very much for your valuable comments. Wei and Hou are students in our group, and Wei did a part of work about Ningxaing pig nutrition and completed his master's thesis, who graduated in 2018. His work is a part of Hou's PhD project, who graduated in 2021. Recent years, we continue to systematically study Ningxiang pig nutrition, meanwhile, Wei started his doctoral study in 2022.  He do the similar study as previous work. Therefore, two papers with the similar title appeared in our description. According to the comments, we have added information in the revised manuscript.